# Transocular sonography in acute arterial occlusions of the eye in elderly patients: Diagnostic value of the spot sign

Michael Czihal[1], Christian Lottspeich[1]*, Anton Köhler[1], Ilaria Prearo[1], Ulrich Hoffmann[1], Siegfried G. Priglinger[2], Marc J. Mackert[2]

1 Division of Vascular Medicine, Medizinische Klinik und Poliklinik IV, Klinikum der Universität München, LMU München, Munich, Germany, 2 Department of Ophthalmology, Klinikum der Universität München, LMU München, Munich, Germany

☯ These authors contributed equally to this work.
* christian.lottspeich@med.uni-muenchen.de

**Data Availability Statement:** All relevant data are within the paper.

**Funding:** The authors received no specific funding for this work.

## Abstract

### Purpose

To characterize the diagnostic yield of the spot sign in the diagnostic workup of acute arterial occlusions of the eye in elderly patients.

### Methods

Clinical characteristics of consecutive patients aged $\geq$ 50 years with acute central retinal artery occlusion (CRAO), branch retinal artery occlusion (BRAO) or anterior ischemic optic neuropathy (AION) were recorded. Videos of transocular sonography were assessed for the presence of the spot sign by two blinded readers. Group comparisons were made between CRAO-patients with and without the spot sign. Two experienced cardiovascular physicians allocated CRAO-cases to a presumed aetiology, without and with knowledge on the presence/absence of the spot sign.

### Results

One-hundred-twenty-three patients were included, 46 of whom suffered from CRAO. A spot sign was seen in 32 of 46 of patients with CRAO and in 7 of 23 patients with BRAO. Interobserver agreement was excellent (Cohen's kappa 0.98). CRAO-patients with the spot sign significantly more frequently had a medical history of cardiovascular disease (62.8 vs. 21.4%, p = 0.03) and left heart valve pathologies (51.9 vs. 10%, p = 0.03). The spot sign was not found in any of the three patients with CRAO secondary to cranial giant cell arteritis. The assumed CRAO aetiology differed in 37% of cases between two cardiovascular physicians, regardless whether transocular sonography findings were known or not.

### Conclusion

The spot sign is a simple sonographic finding with excellent interobserver agreement, which proofs the embolic nature of CRAO, but does not allow exact attribution of the underlying aetiology.

**Competing interests:** The authors have declared that no competing interests exist.

## Introduction

The most common ischemic causes of persistent vision loss are central retinal artery occlusion (CRAO), branch retinal artery occlusion (BRAO), and anterior ischemic optic neuropathy (AION) in elderly patients. These are incisive events for the affected patients, as vision impairment is permanent in most cases. About 80% of patients suffering from CRAO have a final visual acuity of $> 1.3$ on the logarithm of minimum angle resolution (logMAR) scale, defined as blindness by the WHO [1,2].

Giant cell arteritis is an important cause of AION, requiring prompt diagnosis and high-dose corticosteroid treatment [3]. Patients with non-arteritic AION usually share important clinical characteristics with retinal artery occlusions, particularly a high or very high cardiovascular risk profile [4]. While non-arteritic AION is thought to result from a compromised oxygen supply of the optic nerve head due to transient hemodynamic impairment and/or blood desaturation, CRAO and BRAO are characterized by embolic artery occlusions in most cases. The main sources of retinal artery embolism are vessel wall lesions of the supraaortic arteries (mainly arteriosclerosis) and cardiac abnormalities (predominantly originating from the mitral or aortic valve or from the left atrium secondary to atrial fibrillation) [5]. Cases of embolic CRAO/BRAO in association with a patent foramen ovale have been reported [6,7].

Fundoscopy allows direct visualization of the optic nerve head swelling in AION and the emboli in BRAO, whereas the funduscopic findings in CRAO are indirect signs of retinal ischemia. As shown by Schlachetzki et al. in 2010, simple transocular B-mode ultrasound offers visualization of emboli in the central retinal artery by means of a spot-like hyperechoic structure located centrally in the optic nerve [8]. This sonographic finding, entitled "spot sign", has been shown to have a high negative predictive value for cGCA underlying an arteritic CRAO [9]. Moreover, limited data suggest that the spot sign predicts low rates of spontaneous reperfusion and treatment success of systemic thrombolysis [10,11]. In view of the limited evidence available, the purpose of the present cohort study was to characterize the diagnostic yield of the spot sign in addition to the standard diagnostic workup in acute arterial occlusions of the eye in elderly patients.

## Patients and methods

This retrospective study was approved by the local Ethics Committee of the LMU Munich. Data was analyzed retrospectively and anonymously. The following departments/hospitals were involved in the study: Division of Vascular Medicine of the Medical Clinic and Policlinic IV and the Department of Ophthalmology, both parts of the Hospital of the LMU Munich.

Consecutive patients aged $\geq 50$ years with CRAO, BRAO or AION referred for vascular ultrasound between 01/2016 and 12/2019 were included. Patients with only transient visual impairment, chronic symptoms (ocular ischemic syndrome), retinal vein occlusions and non-vascular eye disorders were excluded from the analysis.

The ophthalmological diagnosis was based on fundoscopy, and if appropriate, fluorescein angiography and optical coherence tomography. For the study purposes, the final ophthalmological diagnoses were verified by an ophthalmologist with $> 10$ years professional experience (M.M.). Visual acuity of the affected eye was categorized using the logMAR scale, with higher values indicating worse visual acuity and blindness defined as logMAR $> 1.3$ [2].

Data regarding clinical symptoms, cardiovascular comorbidities, current medication and routine laboratory values were collected. The $CHA_2DS_2$-VASc-score was calculated for all patients. Cardiac examinations (transthoracic and/or transesophageal echocardiography, holter electrocardiogram monitoring) were performed at the discretion of the responsible Internal Medicine specialist. Transthoracic echocardiography examinations were reviewed for the

presence of mitral and aortic valve calcifications/stenoses, hereafter termed "left heart valve pathologies". Transesophageal echocardiography examinations were reviewed with regard to the presence of a patent foramen ovale (PFO) as well as thrombus in the left atrial appendage. For patients with evidence of a PFO the Risk of Paradoxical Embolism (RoPE)-score was calculated [12].

All patients underwent colour duplex sonography (CDS) of the supraaortic arteries and high-resolution sonography of the temporal arteries (LOGIQ E9, GE Healthcare, Milwaukee, Wisconsin, USA). Internal carotid artery stenosis was assessed using a multi-parametric approach, with the degree of stenosis graded according to the North American Symptomatic Carotid Endarterectomy Trial (NASCET) grading system [13]. In addition to the routine diagnostic workup, transocular B-mode sonography of both eyes was performed in all patients using a 9L linear multifrequency transducer (8 MHz). According to current recommendations, the mechanical index was set as low as reasonably achievable (ALARA) in order to limit exposition of the lens and the retina to ultrasound energy [14]. A hyperechoic, sharply delineated structure located centrally within the optic nerve, the optic nerve disc, or the retina nearby the optic nerve disc was considered a positive spot sign (Fig 1) [8,9].

A clinical diagnosis of cGCA was made when at least three of the five following criteria were fulfilled: (1) age > 50 years; (2) typical cranial symptoms (new onset, persisting headache, jaw claudication, temporal artery tenderness); (3) unequivocal symptoms of polymyalgia rheumatica; (4) ESR > 50 mm per 1 hour (reference range ≤ 20 mm per one hour) or C-reactive protein ≥ 2.45 mg/dl; (normal range < 0.5 mg/dl); (5) typical hypoechoic wall thickening (Halo) of the superficial temporal arteries or positive temporal artery biopsy [15].

Group comparisons were made between patients with CRAO, BRAO, and AION as well as between patients with and without the spot sign. The diagnostic accuracy of the spot sign for the exclusion of cGCA was calculated. The mean (SD) distance and diameter of the spot sign were correlated with visual acuity of the affected eye. Transocular sonography videos were independently reviewed by two experienced vascular sonographers with both > 10 years of professional experience (M.C. and C.L.). Sonographic reviewers were blinded to the clinical information in order to assess interobserver agreement.

A specified subgroup analysis focussed on patients with CRAO and compared clinical characteristics of CRAO-patients with and without the spot sign. The diagnostic yield of the spot sign in CRAO-patients with regard to the presumed aetiology was assessed independently by two experienced cardiovascular physicians with > 30 years (U.H.) and > 10 years (A.K.) of professional experience. Aetiology was categorized (embolism of cardiac, arterial or unknown origin; arteritic CRAO in cGCA; CRAO of unknown aetiology) and diagnostic confidence was rated on a four-point Likert scale (4 = excellent, exact diagnosis possible; 3 = good, definite diagnosis possible; 2 = fair, evaluation of major findings possible; and 1 = poor, definite diagnosis impossible). Assessment was repeated with the knowledge of the presence/absence of the spot sign, and rating with and without spot sign-information was compared.

For statistical analysis, SPSS v. 25.0 (SPSS Inc., Chicago, IL, USA) was applied. Univariate group comparisons were performed using Fisher's exact test (categorical variables) and Mann-Whitney-U test (continuous variables). The diagnostic accuracy of the spot sign for the exclusion of a final diagnosis of cGCA was assessed using a 2 x 2 contingency table. Correlation of continuous variables was analysed using Pearson's correlation analysis. For assessment of interobserver agreement, Cohen's kappa was calculated. Two-sided $p$-values <0.05 were considered significant. Results for categorical variables are presented as absolute numbers with percentages, and continuous variables are displayed as mean ± standard deviation (SD).

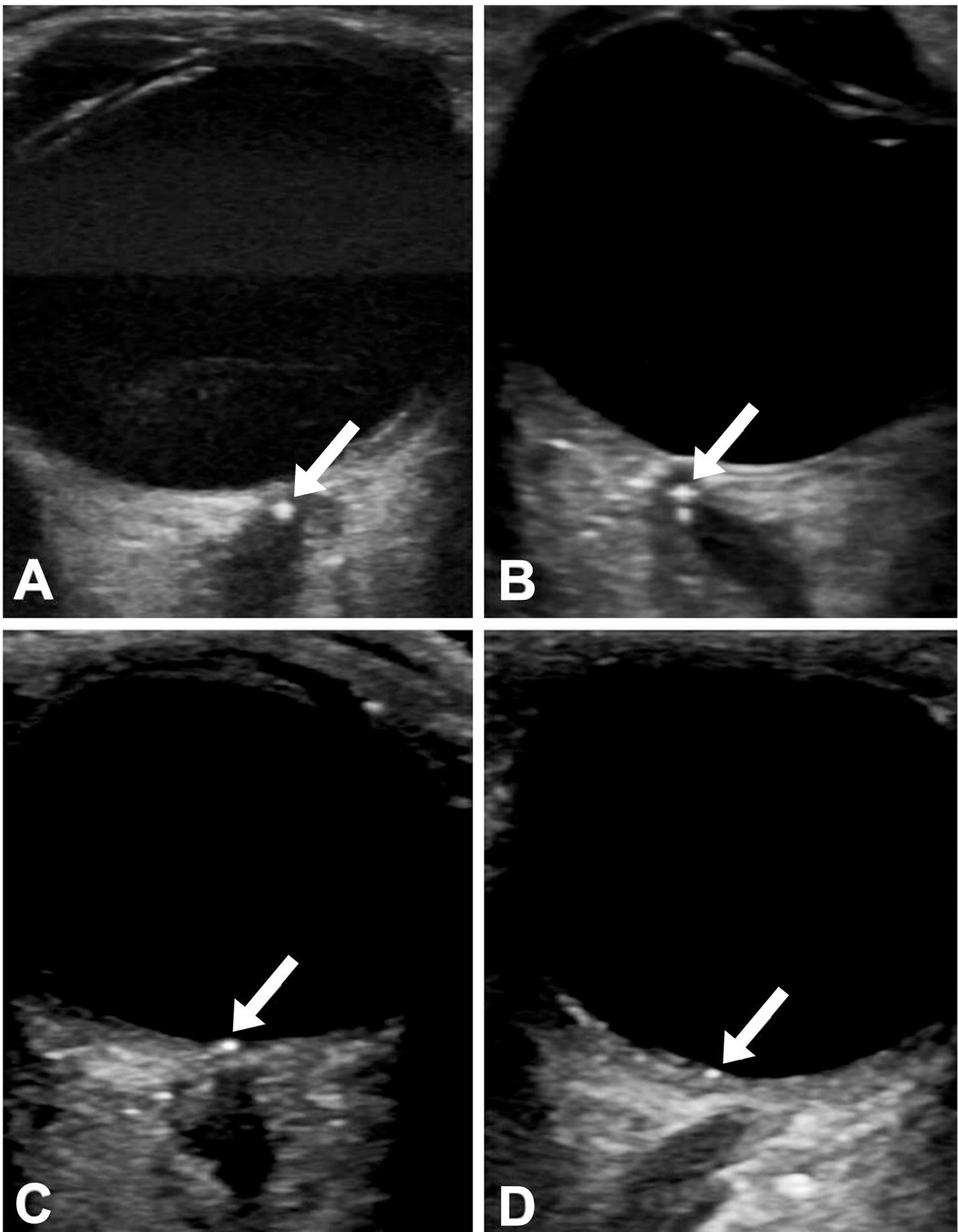

**Fig 1. Spot sign on transocular B-mode ultrasound.** Spot sign at the level of the optic nerve with (A) and without (B) comet tail artefact in CRAO. Spot sign at the level of the optic nerve disc (C) and the retina nearby the optic nerve disc (D) in BRAO is shown.

## Results

### Cohort characteristics

The clinical characteristics of the overall cohort, comprising 123 patients, are summarized in Table 1. The mean age was above 70 years in all subgroups and only 9 patients (10.5%) were younger than 60 years. A final diagnosis of cGCA was established in 50% of patients with AION (n = 27), 6.5% of patients with CRAO (n = 3), and none of the patients with BRAO. All three patients with arteritic CRAO secondary to cGCA suffered from typical cranial symptoms, had elevated inflammatory markers, and typical sonographic findings of the temporal arteries (non-compressible Halo sign).

Noteworthy, BRAO predominantly affected the right eye (69.6%, n = 16/23 vs. left eye 30.4%, n = 7/23), whereas no obvious side predominance was noted in patients with CRAO and AION. Twelve out of 123 patients (9.7%) had bilateral eye involvement. Six of these patients received a final diagnosis of cGCA (5 patients with bilateral AION, one patient with CRAO and contralateral AION) and 4 patients were considered to suffer from bilateral non-arteritic AION. The remaining two patients had metachronous bilateral CRAO and CRAO/BRAO, respectively. Patients with CRAO (n = 46) had worse mean visual acuity compared to the other subgroups (Table 1).

We observed no meaningful clinical differences between patients with CRAO, BRAO and AION, particularly with regard to cardiovascular comorbidities. Patients with CRAO (19.6%,

**Table 1. Clinical characteristics of patients with CRAO, BRAO, and AION in the overall cohort (n = 123).**

| | CRAO (n = 46) | BRAO (n = 23) | AION (n = 54) | p-value |
|---|---|---|---|---|
| Female sex, % | 41.3 | 60.9 | 51.9 | 0.30 |
| Age, years (mean ± SD) | 75.9 ± 9.7 | 73.1 ± 8.7 | 73.4 ± 9.7 | 0.39 |
| Symptom duration, days (mean ± SD) | 6.4 ± 11.9 | 7.7 ± 14.5 | 11.6 ± 28.4 | **0.03** |
| Final diagnosis of cranial GCA, % | 6.5 | 0 | 50 | **< 0.01** |
| Established cardiovascular disease, % | 50 | 39.1 | 29.6 | 0.11 |
| Previous stroke, % | 6.5 | 8.7 | 11.1 | 0.56 |
| Atrial fibrillation, % | 17.0 | 13.0 | 24.1 | 0.45 |
| Active smoking, % | 15.2 | 21.7 | 14.8 | 0.73 |
| Diabetes mellitus, % | 34.8 | 13.0 | 33.3 | 0.14 |
| Arterial hypertension, % | 82.6 | 91.3 | 79.6 | 0.49 |
| Hypercholesterolemia, % | 30.4 | 34.8 | 29.6 | 0.93 |
| Chronic renal insufficiency, % | 30.4 | 30.4 | 31.5 | 1.0 |
| OSAS/COPD, % | 17.4 | 21.7 | 25.9 | 0.60 |
| CHADS2VASC2, mean ± SD | 3.9 ± 1.7 | 3.8 ± 1.7 | 3.2 ± 4.7 | 0.50 |
| Antiplatelet therapy, % | 43.5 | 34.8 | 31.5 | 0.45 |
| Oral anticoagulation, % | 15.2 | 21.7 | 18.5 | 0.75 |
| Statin treatment, % | 37.0 | 26.1 | 24.1 | 0.37 |
| Right eye affected, % | 58.7 | 69.6 | 53.7 | 0.43 |
| Left eye affected, % | 47.8 | 30.4 | 63.0 | **0.01** |
| Both eyes affected, % | 6.5 | 0 | 16.7 | 0.05 |
| Previous amaurosis fugax, % | 6.5 | 0 | 5.6 | 0.74 |
| Previous diplopia, % | 0 | 0 | 7.4 | 0.10 |
| Visual acuity of the affected eye (logMAR), mean ± SD | 1.9 ± 0.9 | 0.7 ± 0.7 | 1.3 ± 1.0 | **< 0.01** |
| Spot sign-positive (on ultrasound), % | 69.6 | 30.4 | 0 | **< 0.01** |

SD, standard deviation.

n = 9/46) and BRAO (17.3%, n = 4/23) more frequently had ipsilateral internal carotid artery stenosis > 50% when compared to patients with AION (3.7%, n = 2/54; p > 0.05). However, ipsilateral high-grade internal carotid artery stenosis > 70% or internal carotid artery occlusion were found in only two patients (high grade carotid artery stenosis in a patient with CRAO who subsequently underwent carotid endarterectomy, total internal artery occlusion in a patient with BRAO). Although the prevalence of atrial fibrillation was substantial in all subgroups (13–24%), a new diagnosis of atrial fibrillation resulting in initiation of oral anticoagulation was made in only 3 patients (two male patients with CRAO, one of whom with a final clinical diagnosis of cGCA; one female patient with bilateral AION secondary to cGCA).

In the group of patients who underwent transthoracic echocardiography (n = 83), left heart valve pathologies were slightly more common in patients with CRAO (40.5%; n = 15/37) compared to patients with BRAO (23.5%; n = 4/17) and AION (24.1%; n = 7/29). High-grade aortic stenosis (aortic valve area < 1 cm$^2$) was found in two patients with CRAO and in one patient with non-arteritic AION. Three patients with CRAO had a history of transcatheter aortic valve implantation (TAVI).

## The spot sign in the overall cohort

None of the patients with AION had sonographic evidence of a spot sign. A spot sign was present in 39 patients (32.0%), including 32 out of 46 patients with CRAO (69.6%) and 7 out of 23 patients with BRAO (30.4%) (Fig 1). Four patients with CRAO who presented on the day of symptom onset (3.3% of the overall cohort) all exhibited a positive spot sign.

The mean distance of the spot sign to the optic nerve disc and the mean diameter of the spot sign were 1.28 ± 0.60 mm and 0.97 ± 0.28 mm in subjects with CRAO, and 0.17 ± 0.20 mm and 0.68 ± 0.10 mm in patients with BRAO, respectively. In the 7 patients with BRAO, the spot sign was located in the optic nerve disc (n = 3, two patients with resulting ischemia of the macula) or the retina (n = 4).

Based on the independent review of B-mode sonography videos of 183 eyes, interobserver agreement for the presence/absence of the spot sign was excellent (agreement in all eyes but one, Cohen's kappa 0.98, p > 0.01). The discrepant case was finally classified as having no spot sign after consensus reading.

## The spot sign in patients with CRAO

We performed further analysis in the subgroup of 46 patients with CRAO. Three of these patients received a final diagnosis of cGCA (including one patient with contralateral AION), and none of them exhibited the spot sign in transocular sonography. On the other hand, none of the patients with a spot sign received a final clinical diagnosis of GCA (negative predictive value of the spot sign for the diagnosis of cGCA 100%).

Comparing spot sign-positive and spot sign-negative patients with CRAO, we found a somewhat shorter time interval between patient-reported symptom onset and the diagnostic examination in spot sign-positive CRAO (4.6 ± 8.1 days) compared to spot sign-negative CRAO (10.5 ± 17.6 days). However, this difference was not statistically significant (p = 0.09). We further observed no significant difference with regard to the mean visual acuity (logMAR 2.1 ± 0.8 vs. 1.6 ± 1.0, p = 0.11). Visual acuity was severely reduced (logMAR > 1.3 of the affected eye) in 84.4% and 71.4% of spot sign-positive and–negative patients with CRAO, respectively (p = 0.42). Neither the diameter of the spot sign (Pearson's rho -0.06) nor the distance of the spot sign to the optic nerve head (rho 0.17) was significantly associated with visual acuity in the 32 patients with spot sign-positive CRAO (p = 0.75 and 0.37, respectively).

There were no significant differences between patients with and without the spot sign regarding age (75.8 ± 8.1 years vs. 76.0 ± 13.0 years, p = 0.78) and cardiovascular risk profile as expressed by the $CHA_2DS_2$-VASc-score (4.2 ± 1.7 vs. 3.4 ± 1.7, p = 0.19). The burden of carotid atherosclerosis also did not differ between both groups. However, patients with the spot sign more frequently had a medical history of cardiovascular disease (62.8 vs. 21.4%, p = 0.03), and these patients also had a significantly higher rate of left heart valve pathologies (51.9 vs. 10%, p = 0.03), mainly of the aortic valve.

Patients without the spot sign were significantly more commonly active smokers (35.7 vs. 6.3%, p = 0.02). Transesophageal echocardiography was performed in 15/46 patients with CRAO (6 spot sign-negative, 9 spot sign-positive). One out of the nine spot sign-positive patients (11%) was diagnosed with a PFO but ipsilateral carotid artery stenosis > 50% was determined as the underlying cause of CRAO in this case. By contrast, five out of the six spot sign negative patients (83%, p < 0.01) had evidence of a PFO, three of whom had a RoPE-score of more than 3 (1 patient with a score of 4 points; two patients with a score of 5 points).

The comparison of sonographic findings in patients with and without spot sign is given in Table 2. Fig 2 shows the relative frequencies of various left heart valve pathologies identified by echocardiography in patients with CRAO.

After independent case review, the assumed CRAO aetiology differed in 37% of cases between two experienced cardiovascular physicians, regardless whether transocular sonography findings (spot sign-positive vs. -negative) were known or not. With the information regarding the presence/absence of the spot sign, both observers more often assigned the cause of CRAO to an arterial embolism source (observer 1: 5 additional cases; observer 2: 10 additional cases, Fig 3). The level of diagnostic confidence significantly increased with the knowledge of transocular sonography findings (observer 1: mean diagnostic confidence level from 2.8 ± 0.7 to 2.1 ± 1.1; observer 2: 2.7 ± 0.7 to 1.9 ± 0.8; p< 0.01). The rate of diagnoses with a diagnostic confidence grade 1 or 2 increased from 34.8% to 54.3% (observer 1) and from 39.1% to 67.4 (observer 2).

## Discussion

In our study on elderly patients with acute arterial occlusions of the eye we found that the spot sign, as depicted by transocular sonography,

i.  is highly specific for the diagnosis of embolic CRAO when located within the optic nerve.

ii.  has a 100% negative predictive value for arteritic CRAO resulting from cGCA, as previously shown by Ertl et al. [9].

iii.  exhibits excellent interobserver agreement (Cohen's kappa 0.98).

**Table 2. Comparison of sonographic findings in patients with and without the presence of the spot sign in the subgroup of patients with CRAO.**

|  | CRAO, spot sign- positive | CRAO, spot sign-negative | p-value |
|---|---|---|---|
| Ipsilateral carotid plaque/stenosis < 50%, n (%) | 24/32 (75) | 11/14 (78.6) | 0.73 |
| Ipsilateral carotid stenosis > 50%, n (%) | 7/32 (21.9) | 2/14 (14.3) | 0.73 |
| Any left heart valve pathology*, n (%) | 14/27 (51.9) | 1/10 (10) | **0.03** |
| Persistent foramen ovale#, n (%) | 1/9 (11.1) | 5/6 (83.3) | **0.01** |
| Atrial fibrillation, n (%) | 5/32 (15.6%) | 3/14 (21.4) | 0.68 |

*determined by transthoracic echocardiography in 37 out of 46 patients.
#determined by transesophageal echocardiography in 15 out of 46 patients.

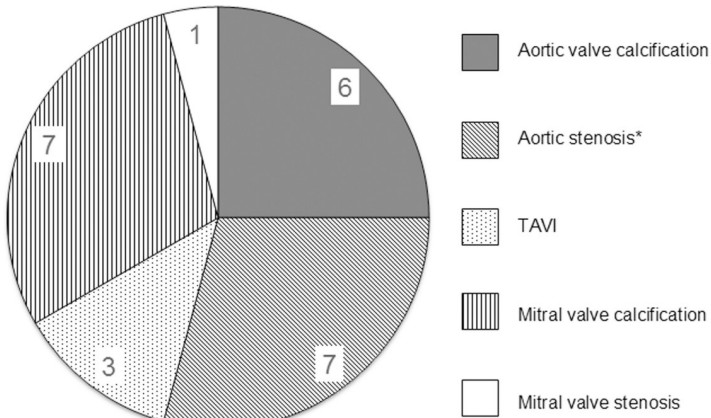

**Fig 2. Types of left heart valve pathologies detected by echocardiography.** Types of left heart valve pathologies detected by echocardiography in patients with CRAO. More than one pathology per patient is possible. *Two patients had high grade aortic stenosis (aortic valve area < 1.0 cm$^2$).

iv.  significantly increases the diagnostic confidence level with regard to the assumed aetiology of CRAO.

v.  does not allow differentiation between an arterial and a cardiac source of embolic CRAO, but may be helpful in some cases guiding the extended diagnostic workup (transesophageal echocardiography in spot sign-negative CRAO).

vi.  may be visualized at the level of the optic nerve disc or the retina nearby in some cases with BRAO.

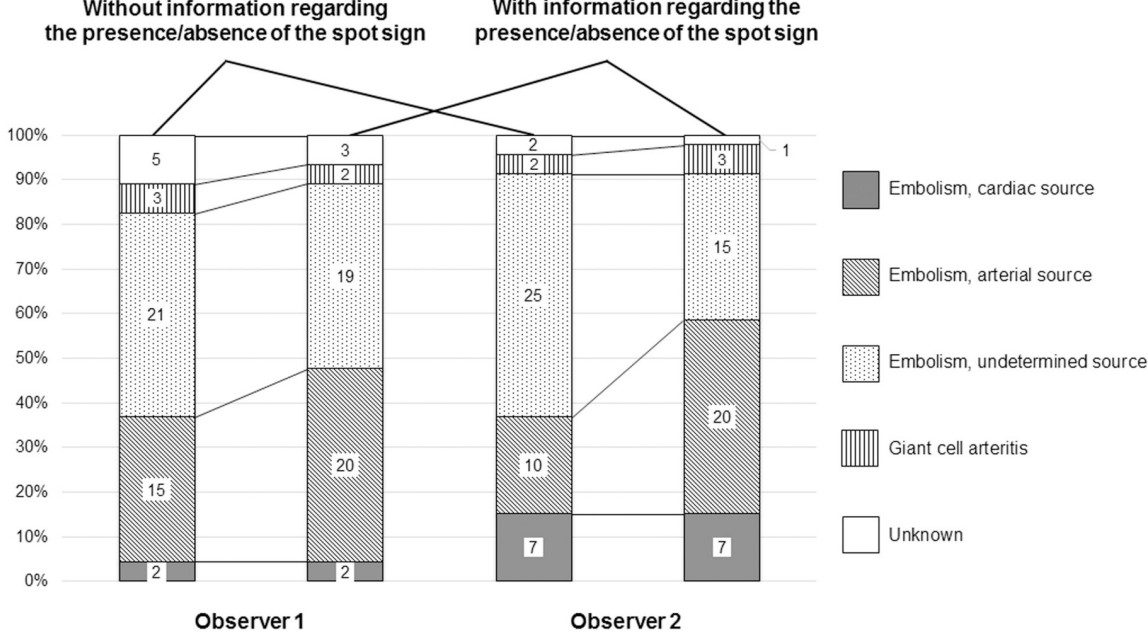

**Fig 3. Aetiology of 46 cases with CRAO.** Assumed aetiology of 46 cases with CRAO after independent case review by two experienced cardiovascular physicians, with and without the knowledge of the presence or absence of a spot sign.

CRAO requires a thorough diagnostic workup, not least because the affected patients are at markedly increased risk of subsequent ischemic stroke [16,17]. Based on the above-mentioned findings, we propose to add simple transocular B-mode sonography to the standard diagnostic workup in CRAO. Transocular B-mode sonography can be performed as a point-of-care test in the emergency department, can be easily learned, is highly reproducible, and requires only a short examination time [18].

The primary aim of transocular sonography is to rule out cGCA as a rare cause of CRAO (about 5% of cases in the literature, 6.5% in our series) based on the presence of the spot sign [1,5]. Noteworthy, all three patients with arteritic CRAO in our series also had marked temporal artery thickening in high-resolution sonography of the temporal arteries (non-compressible Halo sign). However, recently it has been shown that in older patients with high cardiovascular risk, the temporal arteries are sometimes thickened as a result of lifelong exposure to cardiovascular risk factors [19]. The presence of a spot sign proofs the non-arteritic nature of CRAO in such patients. However, regardless of the presence of the spot sign, the diagnostic workup should be as precise as possible in all ocular perfusion disorders.

Beyond the exclusion of cGCA, a positive spot sign may have further diagnostic implications. Nedelmann et al. documented that a spot sign was found more frequently (59%) in patients with embolic CRAO of arterial origin compared to cardio-embolic CRAO (20%) [11]. In our study, there was no difference between spot sign-positive and spot sign-negative CRAO regarding the presence and severity of carotid arteriosclerosis. However, CRAO-patients with a positive spot sign had a significantly higher rate of degenerative pathologies of the aortic and mitral valve. This observation does not necessarily mean that these CRAO were of cardio-embolic origin. It rather reflects advanced calcification/arteriosclerosis of the entire cardiovascular system. This assumption is underlined by the fact that the rate of patients with an established cardiovascular disease was threefold higher in our spot sign-positive CRAO-patients compared to those without a spot sign.

Not only the presence but also the absence of a spot sign may help the clinician to guide the further diagnostic workup, as red thrombi originating from a low flow state such as the left atrial appendix in atrial fibrillation may be suspected in patients with spot sign-negative CRAO [18]. In our series, the rate of newly detected atrial fibrillation was low. In only one patient with spot sign-negative CRAO oral anticoagulation was initiated because of previously unknown atrial fibrillation. However, in the subset of CRAO-patients who underwent transesophageal echocardiography, PFO was found in 5 of 6 patients without a spot sign, compared to 1 out of 9 patients with a spot sign. Three out of 5 spot sign-negative patients with PFO had a RoPE-score > 3, indicating a relevant chance of causal association of the PFO with the CRAO [12]. Therefore, transesophageal echocardiography should be considered in addition to routine diagnostic procedures in patients with spot sign-negative CRAO. Since it is unknown whether or not interventional PFO closure may be beneficial in this particular clinical situation, detection of a PFO will rarely affect patient's management.

Somewhat surprisingly, we found a significantly higher rate of active smokers in patients with spot sign-negative CRAO compared to spot sign-positive CRAO. In these patients, local vasospam in response to serotonine release by activated platelets on arteriosclerotic plaques may play a pathophysiological role [18].

Although most CRAO are assumed to have an arterial source of embolism, ipsilateral high-grade carotid artery stenosis is found in only a minority of these patients. In patients with CRAO and non-stenotic arteriosclerotic lesions of the supraaortic arteries, short time (21 days) dual antiplatelet therapy may be beneficial (analogous to data on patients with minor ischemic stroke and transient ischemic attack) [20]. Whether the antithrombotic treatment

strategy should be influenced by the presence or absence of the spot sign (e.g., antiplatelet therapy vs. oral anticoagulation) is unclear.

Thrombolysis may be beneficial in CRAO with symptom duration of less than 4.5 hours, and the spot sign has been shown to negatively predict treatment success of thrombolysis [11]. In our series, only 4 patients presented with symptom duration < 24 hours, all of whom had sonographic evidence of the spot sign. Therefore, in our opinion thrombolysis should be reserved for carefully selected cases (young patient, very short symptom duration, negative spot sign).

Some limitations of our study warrant consideration. First, study data were collected retrospectively. Second, the cardiological workup as well as the neurological workup was incomplete in some cases. Third, Doppler examinations of the central retinal arteries were not routinely performed and routine imaging of the aortic arch and the intracranial arteries was not part of the diagnostic workup. Therefore, the potential impact of arteriosclerotic lesions in these arterial territories (the intracranial internal carotid artery has been shown previously to be frequently affected by arteriosclerosis in CRAO [21]) remains unclear.

Finally, we did not prospectively follow up the patients. Thus we cannot provide information on the clinical course (visual acuity, cardiovascular events) of spot sign-positive and negative CRAO-patients as well as of patients with BRAO and AION. The high rate of patients with arteritic AION (50% in our series, compared to 5–10% in the literature [22]) may be explained by a selection bias (patients with unequivocal non-arteritic AION are not routinely referred for vascular ultrasound).

In summary, our study confirmed the high negative predictive value of the spot sign for exclusion of cGCA in patients with acute arterial occlusion of the eye. We further showed an excellent interobserver-agreement of this easily obtainable sonographic finding. Moreover, our results suggest that presence or absence of a spot sign does not provide exact aetiologic information on CRAO but helps to guide the further diagnostic workup. Knowledge regarding the presence or absence of a spot sign seems to be reassuring with regard to the assumed aetiology of CRAO.

## Author Contributions

**Conceptualization:** Michael Czihal, Christian Lottspeich, Ulrich Hoffmann, Marc J. Mackert.

**Data curation:** Michael Czihal.

**Formal analysis:** Michael Czihal, Christian Lottspeich, Anton Köhler, Ulrich Hoffmann, Marc J. Mackert.

**Investigation:** Michael Czihal, Christian Lottspeich, Anton Köhler, Ilaria Prearo, Marc J. Mackert.

**Methodology:** Michael Czihal, Christian Lottspeich, Anton Köhler, Ulrich Hoffmann, Siegfried G. Priglinger, Marc J. Mackert.

**Resources:** Michael Czihal, Marc J. Mackert.

**Supervision:** Michael Czihal, Ulrich Hoffmann, Siegfried G. Priglinger, Marc J. Mackert.

**Validation:** Michael Czihal, Christian Lottspeich, Anton Köhler, Ilaria Prearo, Ulrich Hoffmann.

**Visualization:** Christian Lottspeich.

**Writing – original draft:** Michael Czihal, Christian Lottspeich.

**Writing – review & editing:** Michael Czihal, Christian Lottspeich, Anton Köhler, Ilaria Prearo, Ulrich Hoffmann, Siegfried G. Priglinger, Marc J. Mackert.

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
