## [Decision Letter · Decision Letter 0]

14 Jan 2021

PONE-D-20-38731

Transocular sonography in acute arterial occlusions of the eye in elderly patients: diagnostic value of the spot sign

PLOS ONE

Dear Dr. Lottspeich,

Thank you for submitting your manuscript to PLOS ONE. After careful consideration, we feel that it has merit but does not fully meet PLOS ONE’s publication criteria as it currently stands. Therefore, we invite you to submit a revised version of the manuscript that addresses the points raised during the review process.

We look forward to receiving your revised manuscript.

Kind regards,

Aristeidis H. Katsanos, MD, PhD

Academic Editor

PLOS ONE

Journal Requirements:

2. Please include in your Methods section (or in Supplementary Information files) the participating hospitals/institutions.

4. Thank you for submitting the above manuscript to PLOS ONE. During our internal evaluation of the manuscript, we found that your abstract was previously published as part of the following conference proceedings: 4. DGA-Interventionskongress und 49. Jahrestagung Deutsche Gesellschaft für Angiologie – Gesellschaft für Gefäßmedizin e. V. (https://econtent.hogrefe.com/doi/10.1024/0301-1526/a000866).

Before we proceed, would you please kindly clarify if the published extended abstract was previously peer-reviewed? If so, please explain in your cover letter why this work does not constitute a dual publication.

Would you please provide the following information, as well:

1) Please explain how your submission builds on the published extended abstract in Vasa (European Journal of Vascular Medicine) and whether any additional information or data are presented in your PLOS ONE submission, that were not presented in the Vasa extended abstract.

2) Please note that should your paper be accepted, all content including images will be published under the Creative Commons Attribution (CC BY) 4.0 license, which means that they will be freely available online, and any third party is permitted to access, download, copy, distribute, and use these materials in any way, even commercially, with proper attribution. In order to publish any previously copyrighted material, PLOS ONE requires permission from the original copyright holder of the content to publish it under the CC BY 4.0 license.

Please clarify whether the authors have received written permission from Vasa to publish this content specifically under the CC BY 4.0 license and upload the granted permission to the manuscript as a supporting information file.

Please note that further consideration is dependent on the submission of a manuscript that addresses these concerns about the overlap in text with published work.

Reviewers' comments:

Reviewer's Responses to Questions

**Comments to the Author**

1. Is the manuscript technically sound, and do the data support the conclusions?

Reviewer #1: Yes

Reviewer #2: Yes

2. Has the statistical analysis been performed appropriately and rigorously? 

Reviewer #1: Yes

Reviewer #2: Yes

3. Have the authors made all data underlying the findings in their manuscript fully available?

Reviewer #1: Yes

Reviewer #2: Yes

4. Is the manuscript presented in an intelligible fashion and written in standard English?

Reviewer #1: Yes

Reviewer #2: Yes

5. Review Comments to the Author

Reviewer #1: This is a well performed and helpful retrospective study, evaluating the prevalence and diagnostic value of the sonographic retrobulbar spot sign in patients with different etiologies of acute arterial occlusions fo the eye.

All the five reported findings/conclusion (some confirmatory, some new) are of clinical and scientific interest.

The methods are sound. The paper is well written.

I have only the following minor comments:

- Introduction, second paragraph: please include atrial fibrillation as one possible source of cardioembolism.

- Results: also the absolute numbers should be given in the text, not only the percentage

- page 9, lines 204-208: the difference in mean distance of the spot sign between CRAO and BRAO is an important finding and of clinical interest. This should be also included in the discussion as one of the main findings of the study (resulting in a total of 6 findings)

- when reporting about RoPE-Score results, it would be more of interest to report the number of patients with a RoPE-Score >5 than >3.

- did the patients also received neurological consultation?

- Did the authors also performed doppler-examination of the ocular arteries and veins? If not, this should be included in the limitation section.

- Table 1: a fourth column should be included, with p-values of comparison of the three subgroubs

- Table 1: Also, a furhter row sould be included, indicating the number (prevalence) of spot-sing-positive patients in each subgroup

- Tabel 2: a further row should be included, indicatin the number (prevanlence) of atrial fibrillation

Reviewer #2: I thank the editor Prof. Emily Chenette for inviting me to review the manuscript:Transocular sonography in acute arterial occlusions of the eye in elderly patients: diagnostic value of the spot sign --Manuscript Draft-- Manuscript Number: PONE-D-20-38731

Summary: This is an interesting observational retrospective study exploring the uselfulness of transocular sonography to detect the spot sign in a large cohort of patients. One-hundred-twenty-three with acute arterial occlusions of the eye in elderly patients were evaluated. A spot sign was seen in 32 of 46 of patients with CRAO and in 7 of 23 patients with BRAO.

Moreover the spot sign was encountered as a simple sonographic finding with excellent interobserver agreement by two different authors.

Methods:Adequate sample size.

Results: adequate.

Minor points- Discussion:

Some of the considerations in the discussion are a little questionable, given the retrospective nature of the study. First although some patients without the spot sign present PFO (5 of 6 patients without a spot sign), compared to 1 out of 9 patients with a spot sign, I don’t believe that the presence of a PFO in an elderly population will change the management of patients.

Second only the absence of spot sign in the clinical contest with criteria of cGCA helps clinicians, to orientate the diagnostic process. Regardless of the presence of the of spot sign in CRAO or BRAO (embolic of arterial or cardio-embolic origin), the search for the cause should the most accurate as possible

Finally I would suggest to add, if it is possible, an interesting information about latency of onset of symptoms and the time of examination

Conclusion:

Major finding is the high frequency to detect ultrasonographically. the spot sign in a large cohort of patients with CRAO> BRAO. Second the level of diagnostic confidence of aetiology seems to be increased with knowledge of transorbital sonography findings, third the confirmation of absence of spot sign in patients affected by cGCA.

Considering that this technique is a simple, safe, bedside and noninvasive and provides the advantage of a rapid work-up of patients.

Reference were relevant literature

Graphic illustrationsI adequate.

6. PLOS authors have the option to publish the peer review history of their article (what does this mean?). If published, this will include your full peer review and any attached files.

Reviewer #1: No

Reviewer #2: No

---

## [Author Response · Author response to Decision Letter 0]

29 Jan 2021

Point-by-point-Response 

(The line numbers given refer to the marked up version of the manuscript)

Editors Comments

1) “Please ensure that your manuscript meets PLOS ONE's style requirements, including those for file naming. The PLOS ONE style templates can be found at […]”

We have ensured that the manuscript meets PLOS ONE's style requirements. The manuscript and file naming (figures) was adapted accordingly. Figure legends have been moved from the appendix section to the text. 

2) “Please include in your Methods section (or in Supplementary Information files) the participating hospitals/institutions.”

A corresponding sentence was added to the method section (lines 94ff). 

3) “We note that you have included the phrase “data not shown” in your manuscript. Unfortunately, this does not meet our data sharing requirements. PLOS does not permit references to inaccessible data. We require that authors provide all relevant data within the paper, Supporting Information files, or in an acceptable, public repository. Please add a citation to support this phrase or upload the data that corresponds with these findings to a stable repository (such as Figshare or Dryad) and provide and URLs, DOIs, or accession numbers that may be used to access these data. Or, if the data are not a core part of the research being presented in your study, we ask that you remove the phrase that refers to these data.”

In the revised paper, the requested data has now been included (lines 251ff).

4) Thank you for submitting the above manuscript to PLOS ONE. During our internal evaluation of the manuscript, we found that your abstract was previously published as part of the following conference proceedings: 4. DGA-Interventionskongress und 49. Jahrestagung Deutsche Gesellschaft für Angiologie – Gesellschaft für Gefäßmedizin e. V. (https://econtent.hogrefe.com/doi/10.1024/0301-1526/a000866).

• Before we proceed, would you please kindly clarify if the published extended abstract was previously peer-reviewed? If so, please explain in your cover letter why this work does not constitute a dual publication. Would you please provide the following information, as well:

 1) Please explain how your submission builds on the published extended abstract in Vasa (European Journal of Vascular Medicine) and whether any additional information or data are presented in your PLOS ONE submission, that were not presented in the Vasa extended abstract.

2) Please note that should your paper be accepted, all content including images will be published under the Creative Commons Attribution (CC BY) 4.0 license, which means that they will be freely available online, and any third party is permitted to access, download, copy, distribute, and use these materials in any way, even commercially, with proper attribution. In order to publish any previously copyrighted material, PLOS ONE requires permission from the original copyright holder of the content to publish it under the CC BY 4.0 license.

• Please clarify whether the authors have received written permission from Vasa to publish this content specifically under the CC BY 4.0 license and upload the granted permission to the manuscript as a supporting information file.

• Please note that further consideration is dependent on the submission of a manuscript that addresses these concerns about the overlap in text with published work. 

Indeed, this work has been presented previously (oral presentation) at the 49th annual meeting of the Deutsche Gesellschaft für Angiologie – Gesellschaft für Gefäßmedizin e. V. (09/2020). The abstract of this oral presentation has been published within the conference proceedings in VASA 2020 (Volume 49 / Supplement 105 / 2020), as usually done for such conferences. We clearly line out the previous presentation of our preliminary results as conference abstract in our revised paper. We hereby assure that our work has not been peer-reviewed previously and is not under consideration for publication elsewhere. There is no previously copyrighted material. Our full paper submitted to PLOS ONE contains considerably more information than orally presented as conference abstract. These points underscore the present work does not present a dual publication. 

Attached please find a letter from the Hogrefe AG (publisher of VASA) confirming the above mentioned statement and granting full permission to PLOS One to publish all content under the CC BY 4.0 licence (Response_To_Reviewers_supplement.pdf). 

Reviewer Comments to Author

Reviewer #1

1) “This is a well performed and helpful retrospective study, evaluating the prevalence and diagnostic value of the sonographic retrobulbar spot sign in patients with different etiologies of acute arterial occlusions for the eye. All the five reported findings/conclusion (some confirmatory, some new) are of clinical and scientific interest. The methods are sound. The paper is well written. I have only the following minor comments: Introduction, second paragraph: please include atrial fibrillation as one possible source of cardioembolism. Results: also the absolute numbers should be given in the text, not only the percentage”

Thank you for your comments. Atrial fibrillation has been added in the introduction paragraph (line 77) and in the results section absolute numbers of patients were added (lines 182, 192f, 202ff, 215ff).

2) “page 9, lines 204-208: the difference in mean distance of the spot sign between CRAO and BRAO is an important finding and of clinical interest. This should be also included in the discussion as one of the main findings of the study (resulting in a total of 6 findings)”. “when reporting about RoPE-Score results, it would be more of interest to report the number of patients with a RoPE-Score >5 than >3.”

We completely agree and added the requested information in the results section (lines 269f). Furthermore, we added a 6th major conclusion to the discussion (line 316f).

3) “- did the patients also received neurological consultation?”. “Did the authors also performed doppler-examination of the ocular arteries and veins? If not, this should be included in the limitation section.”

Both doppler examinations of the eyes and structured neurologic exams were only partly done in some patients but not in a standardized fashion, therefore this previously has not been included in the manuscript. We added a passage in the Discussion (lines 381ff). 

4) “- Table 1: a fourth column should be included, with p-values of comparison of the three subgroups“

“- Table 1: Also, a further row should be included, indicating the number (prevalence) of spot-sing-positive patients in each subgroup”

”- Table 2: a further row should be included, indicating the number (prevalence) of atrial fibrillation”

 All additions to the tables were made accordingly (lines 188f; lines 275f). 

Reviewer #2

1) “I thank the editor Prof. Emily Chenette for inviting me to review the manuscript: Transocular sonography in acute arterial occlusions of the eye in elderly patients: diagnostic value of the spot sign --Manuscript Draft-- Manuscript Number: PONE-D-20-38731. Summary: This is an interesting observational retrospective study exploring the usefulness of transocular sonography to detect the spot sign in a large cohort of patients. One-hundred-twenty-three with acute arterial occlusions of the eye in elderly patients were evaluated. A spot sign was seen in 32 of 46 of patients with CRAO and in 7 of 23 patients with BRAO. Moreover the spot sign was encountered as a simple sonographic finding with excellent interobserver agreement by two different authors. Methods: Adequate sample size. Results: adequate. Minor points: - Discussion: Some of the considerations in the discussion are a little questionable, given the retrospective nature of the study. First although some patients without the spot sign present PFO (5 of 6 patients without a spot sign), compared to 1 out of 9 patients with a spot sign, I don’t believe that the presence of a PFO in an elderly population will change the management of patients.”

Thank you for your comment. In the discussion we already mentioned the retrospective nature of our study as an important limitation. We agree that the detection of a PFO will rarely change the management of these patients and changed our commentary regarding this issue (lines 357ff).

2) “Second only the absence of spot sign in the clinical contest with criteria of cGCA helps clinicians, to orientate the diagnostic process. Regardless of the presence of the spot sign in CRAO or BRAO (embolic of arterial or cardio-embolic origin), the search for the cause should the most accurate as possible.”

We completely agree that the search of the source of embolism should be the clinicians’ prior task and added a sentence to the discussion (lines 333f).

3) “Finally I would suggest to add, if it is possible, an interesting information about latency of onset of symptoms and the time of examination.”

We added a comparison on the diagnostic latency between spot-positive and negative patients in the results section (lines 244ff). 

4) “Conclusion: Major finding is the high frequency to detect ultrasonographically. the spot sign in a large cohort of patients with CRAO>BRAO. Second the level of diagnostic confidence of aetiology seems to be increased with knowledge of transorbital sonography findings, third the confirmation of absence of spot sign in patients affected by cGCA. Considering that this technique is a simple, safe, bedside and noninvasive and provides the advantage of a rapid work-up of patients. Reference were relevant literature. Graphic illustrations adequate.”

Thanks again to the editors and both reviewers for your time and comments in the review process. We are looking forward to your further feedback and Editorial decision. 

Best regards,

Christian Lottspeich

(on behalf of all authors)

---

## [Editor Report · Decision Letter 1]

1 Feb 2021

Transocular sonography in acute arterial occlusions of the eye in elderly patients: diagnostic value of the spot sign

PONE-D-20-38731R1

Dear Dr. Lottspeich,

We’re pleased to inform you that your manuscript has been judged scientifically suitable for publication and will be formally accepted for publication once it meets all outstanding technical requirements.

Kind regards,

Aristeidis H. Katsanos, MD, PhD

Academic Editor

PLOS ONE
---

## [Editor Report · Acceptance letter]

4 Feb 2021

PONE-D-20-38731R1 

Transocular sonography in acute arterial occlusions of the eye in elderly patients: diagnostic value of the spot sign 

Dear Dr. Lottspeich:

I'm pleased to inform you that your manuscript has been deemed suitable for publication in PLOS ONE. Congratulations! Your manuscript is now with our production department. 

Kind regards, 

on behalf of

Dr. Aristeidis H. Katsanos 

Academic Editor

PLOS ONE